# Reward Translation via Reward Machine in Semi-Alignable MDPs

## Abstract

Deep reinforcement learning often relies heavily on the quality of dense rewards, which can necessitate significant engineering effort. Reusing human-designed rewards across similar tasks in different domains can enhance learning efficiency in reinforcement learning. Current works have delved into an assortment of domains characterized by divergent embodiments, differing viewpoints, and dynamic disparities. However, these studies require either alignment or alignable demonstrations in which states maintain a bijection, consequently restricting the applicability to more generalized reward reusing across disparate domains. It becomes crucial to identify the latent structural similarities through coarser-grained alignments between distinct domains, as this enables a reinforcement learning agent to harness its capacity for abstract transfer in a manner akin to human navigation based on maps. To address this challenge, semi-alignable Markov Decision Processes (MDPs) is introduced as a fundamental underpinning to delineate the coarse-grained latent structural resemblances amidst varying domains Subsequently, the Neural Reward Translation (NRT) framework is established, which employs reward machines to resolve cross-domain reward transfer problem within semi-alignable MDPs, thus facilitating more versatile reward reusing that supports reinforcement learning across diverse domains. Our methodology is corroborated through several semi-alignable environments, highlighting NRT's efficacy in domain adaptation undertakings involving semi-alignable MDPs.

## 1 Introduction

Deep reinforcement learning (RL) has achieved great success in games (Mnih et al., 2015; Lample & Chaplot, 2017) and practical areas like robotic control (Kober et al., 2013), automatic driving (Zhu et al., 2018), precision agriculture management (Li et al., 2021) etc. However, the effectiveness of current deep reinforcement learning methods is over correlated to the quality of the reward signals. In practical applications, designing dense rewards usually requires significant engineering effort (Fickinger et al., 2021), and human-provided rewards tend to be Non-Markovian (MacGlashan et al., 2017), which can hinder the training process.

Nevertheless, humans excel at recognizing latent structural similarities between tasks in related but distinct domains and abstracting skills from these differences. We can learn from third-person observations—those having no explicit correspondence to our internal self-representations (Stadie et al., 2017; Liu et al., 2018; Sermanet et al., 2018)—such as finding the path on a map and navigating to our destination in real life or imitating experts with different embodiments (Gupta et al., 2017; Rizzolatti & Craighero, 2004; Liu et al., 2019) in foreign environments (Liu et al., 2019). Endowing RL agents with the ability to abstract skills and draw inferences from one domain to reuse the reward in another will enhance learning efficiency.

Efforts for reward reusing across various domains have been ongoing for years but remain limited. Recent work has primarily focused on cross-domain imitation learning, where algorithms learn mappings of observation and action space between expert and agent domains from demonstrations. Existing methods typically rely on three key domain descriptors: dynamics (Liu et al., 2019), embodiment (Gupta et al., 2017; Hudson et al., 2021), and viewpoint mismatch (Jiang et al., 2020; Stadie et al., 2017). However, these preliminary methods depend on paired, time-aligned demonstrations and can handle only one descriptor at a time. More recent work relaxes these constraints

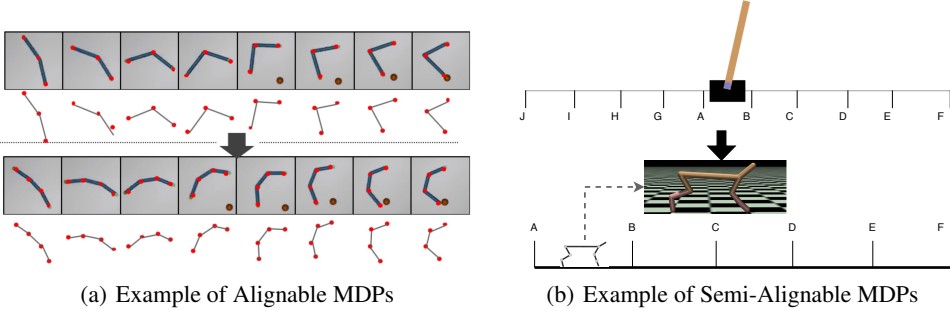

(a) Example of Alignable MDPs     (b) Example of Semi-Alignable MDPs

Figure 1: Example of Semi-Alignable MDPs

by requiring only alignable expert demonstrations with various alignment self-discovery schemes, enabling algorithms to automatically learn observation and action mappings from unpaired and unaligned demonstrations. For example, DAIL (Kim et al., 2020) learns cross-domain transformation from unpaired or unaligned demonstrations with state and action using Generative Adversarial MDP Alignment and can handle all three descriptors. Raychaudhuri et al. (2021) introduce a CycleGAN technique to learn transformations from unpaired and unaligned demonstrations containing only expert domain states, while Fickinger et al. (2021) use the Gromov-Wasserstein distance to eliminate the need for proxy tasks. Despite advancements in cross-domain imitation learning and reward signal reuse, these methods struggle with more general cross-domain tasks. For instance, they cannot align tasks like Cartpole and HalfCheetah as shown in Figure 1(b). As they have different horizon length, different and unalignable state and action space. Humans, however, can mentally build the relation between the tasks by identifying common checkpoints by abstract thinking.

In this paper, we introduce *semi-alignable MDPs*, where an abstract alignment exists between tasks, as illustrated in Figure 2. By further abstracting tasks to focus on skills and sub-tasks, semi-alignable MDPs allow high-level mapping across domains, more similar to human learning processes. Our goal is to establish a framework that learns these high-level mappings within more generalized semi-alignable MDPs and reuses rewards to enhance training efficiency in new domains. To achieve this, we must abstract tasks for obtaining alignable skill and abstract spaces. Reward Machines (RMs) provide a bridge to uncover task structures with non-Markovian reward functions using high-level events and abstract checkpoints (Icarte et al., 2022). We propose the Neural Reward Translation (NRT) method to solve the reward transfer problem within semi-alignable MDPs. NRT leverages RMs to distill abstract alignments and transfer reward signals between domains.

The primary contributions of this paper are as follows:

1) The definition of semi-alignable MDPs is first introduced to theoretically support more general domain knowledge transfer than classical alignable MDPs in cross-domain reinforcement learning;

2) Based on semi-alignable MDPs and reward machine, a novel framework called Neural Reward Translation is established to solve reward transfer problem in more general cross domain setting within semi-alignable MDPs.

3) Serval semi-alignable environment is proposed to show the performance of Neural Reward Translation in cross domain transferring learning task with environments under semi-alignment MDPs.

Moreover, we address the significant human engineering efforts involved in creating hand-crafted reward machines by integrating an LLM-based framework to construct them, leveraging domain knowledge from resources such as task manuals, as detailed in the Appendix.

## 2    RELATED WORKS

### 2.1    DOMAIN TRANSFER IN REINFORCEMENT LEARNING

Various works have been attempted in transfer learning in the reinforcement learning area (Taylor & Stone, 2009; Zhu et al., 2020). To deal with transferring learning between different domains, primitive methods always try to use the hand-craft features along with a distance metric between

the imitation agent and the expert. For example, Ammar & Taylor (2011) defines a common state space between MDPs from the two domains and uses it to learn a map between the two domains' states. Ammar et al. (2015) uses unsupervised manifold alignment to learn a linear map between states with similar local geometric properties.

Recent works divide the domain transfer into three types: dynamics (Liu et al., 2019), embodiment (Gupta et al., 2017; Hudson et al., 2021), and viewpoint mismatch (Jiang et al., 2020; Stadie et al., 2017). Generally, they obtain states corresponding to the proxy tasks and paired, time-aligned demonstrations and use them to learn a state map or state encoder by deep learning methods. DAIL (Kim et al., 2020) proposed a universal framework for all three types and used GAMA to self-learn the alignment between MDPs in different domains. Furthermore, xDIO (Raychaudhuri et al., 2021) uses a CycleGAN to learn the alignment with state-only demonstrations. GWIL (Fickinger et al., 2021) uses the Gromov-Wasserstein distance to eliminate the need for proxy tasks.

Current domain transfer methods all consider the MDPs which are alignable. However, alignable MDPs from two different domains are still luxurious in real-world applications. In this paper, we aim to distill the abstract alignment from semi-alignable expert demonstrations and transfer it to the imitation agent by an additional reward signal. In this way, we don't ever need the alignable MDPs, which will bring a more general domain transfer in the reinforcement learning area.

## 2.2 REWARD MACHINE

Reward machine is a kind of finite state machine first introduced by Icarte, Rodrigo Toro, et al (Icarte et al., 2018), which is established to reveal the structure of non-Markovian reward functions of tasks that are encoded with high-level events(i.e., propositional variables). Icarte et al. (2018) also combined Q-learning and reward machine, proposed the first reinforcement leanring method within reward machine, QRM. Later, Icarte et al. (2022) proposed counterfactual experiences for reward machines(CRM), a modified version of QRM which learns one Q-function taking reward machine states as a part of the inputs and is more suitable when combined with deep neural network. Icarte et al. (2022) proposed hierarchical reinforcement learning for reward machines (HRM), which can be effective at quickly learning good policies for reinforcement learning task, but might converge to sub-optimal solutions. Icarte et al. (2023) propose a discrete optimization problem for learning reward machines from experience in a partially observable environment.

Additionally, reward machine has been used for solving problems in robotics (Camacho et al., 2021; DeFazio & Zhang, 2021; Shah et al., 2020), multi-agent reinforcement learning (Neary et al., 2021), lifelong reinforcement learning (Zheng et al., 2022) and offline reinforcement leanring (Sun & Wu, 2023). Unlike these works, our paper aims to use reward machine to uncover reinforcement learning task structure and try to transfer the reward signal between tasks with different domains.

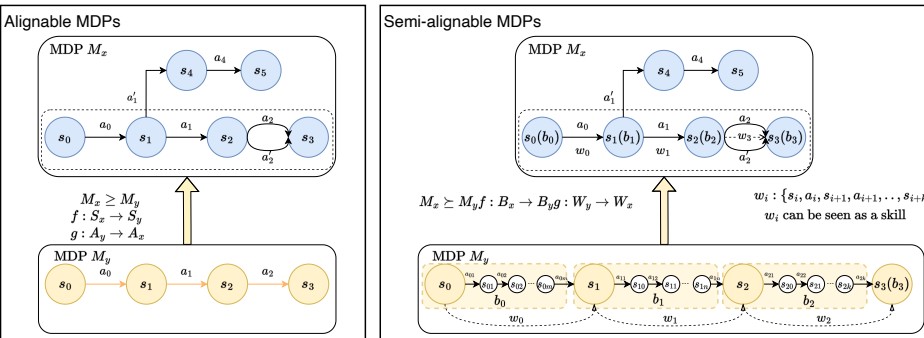

Figure 2: The difference between alignable MDPs and semi-alignable MDPs. In alignable MDPs, there exists injections between state space $S$ and action space $x$ within $M_x$ and $M_y$. For Semi-alignable MDPs, there exists injections between abstract state $y$ space and skill space $W$ within $M_x$ and $M_y$. (The symbol "$\geq$" means an MDP reduction from $M_x$ to $M_y$, while the symbol "$\succeq$" means an MDP semi-reduction from $M_x$ to $M_y$.)

## 3 SEMI-ALIGNABLE MDPS

Before introducing the Neural Reward Translation. We first lay the definitions and properties on semi-alignable MDPs. To help describe the high-level action and state ('skill' and 'goal') in the semi-alignable MDPs shown in Figure. 2. We extend the infinite horizon Markov Decision Process (MDP) $\mathcal{M}$ to let it be expressed as a tuple $(S, x, y, W, P, r, P)$ where $S$ is the state space, $x$ is the action space, $P$ is the transition function $r$ is the reward function. $y$ and $W$ are extended by us, where $y$ is the abstract state space and $W$ is the skill space. Generally, a domain fully characterizes the embodied agent and the environment dynamics, but without desired behaviors. By this way, we define an MDP in one domain $\mathcal{X}$ for a task $\mathcal{T}$ as $\mathcal{M}_x^{\mathcal{T}} = (S_x, A_x, B_x, W_x, P_x, r_x^{\mathcal{T}}, P_x)$. Likewise, an MDP in another domain $\mathcal{Y}$ for the proxy task can be define as $\mathcal{M}_y^{\mathcal{T}} = (S_y, A_y, B_y, W_y P_y, r_y^{\mathcal{T}}, P_y)$. (Kim et al., 2020) has proposed *MDPs alignability theory* and *MDP reduction* where an MDP reduction from $M_x$ to $M_y$ holds a tuple which maps preserve dynamics, which means for any $(s_x, a_x, s_y, a_y) \in \mathcal{S}_x \times \S_x \times \mathcal{S}_y \times \S_y$, there exists an $\phi$, which let $P_y(s_y, a_y) = \phi(P_x(s_x, a_x))$. We further define the *MDP semi-reduction* and *semi-alignable MDPs*.

**Definition 1.** *An **MDP semi-reduction** from $\mathcal{M}_x^{\mathcal{T}} = (S_x, A_x, B_x, W_x, P_x, r_x^{\mathcal{T}}, P_x)$ to $\mathcal{M}_y^{\mathcal{T}} = (S_y, A_y, B_y, W_y P_y, r_y^{\mathcal{T}}, P_y)$ is a tuple $r = (\phi, \psi)$ where $\phi : B_x \to B_y$, $\psi : W_x \to W_y$, which preserve:*

- *$\pi^w$-optimality: $\forall (b_x, w_x, b_y, w_y) \in B_x \times W_x \times B_y \times W_y$:*

$$O_{M_y}(\phi(b_x), \psi(w_x)) = 1 \Rightarrow O_{M_x}(b_x, w_x) = 1,$$

$$O_{M_y}(b_y, w_y) = 1 \Rightarrow \phi^{-1}(b_y) \neq \emptyset, \psi^{-1}(w_y) \neq \emptyset.$$

- *$y$-dynamic: $\forall (b_x, w_x, b_y, w_y) \in B_x \times W_x \times B_y \times W_y$ such that $O_{M_y}(b_y, w_y) = 1, b_x \in \phi^{-1}(b_y), w_x \in \phi^{-1}(w_y)$:*

$$P_y^B(b_y, w_y) = \phi(P_x^B(b_x, w_x)).$$

This definition is a direct expansion of the MDP reduction. Following it, we can get the definition of the semi-alignable:

**Definition 2.** *Two MDPs $M_x$, $M_y$ are semi-alignable if and only if $M_x \succeq M_y$ or $M_y \succeq M_x$, where $M_x$ and $M_y$ are any two MDPs in domain $\mathcal{X}$ and domain $\mathcal{Y}$ while $M_x \succeq M_y$ means there is a tuple $(\phi, \psi)$ which can make the semi-reduction from $M_x$ to $M_y$.*

However, direct finding the semi-reduction between $\mathcal{M}_x^{\mathcal{T}}$ and $\mathcal{M}_y^{\mathcal{T}}$ because the abstract state space $y$ and the skill space $W$ in both domains are indeterminate. In this paper, *Neural Reward Translation* (NRT) uses *Reward Machine (RM)* to indirectly build the relationship on MDPs within different domains, then transfer the reward from one task to another to improve the training efficiency.

## 4 NEURAL REWARD TRANSLATION

In this section, we introduce the primary framework called *Neural Reward Translation* (NRT), whose architecture is depicted in Figure 3. The NRT approach facilitates the generation of corresponding reward machines through human-defined systems or large language models based on task manuals. In this paper, we mainly focus on utilizing a human-defined reward machine. But we also propose an LLM-based framework based on chain of thought (Wei et al., 2022), using GPT-4 (OpenAI, 2023) to generate reward machines from task manuals. This framework demonstrates how state-of-the-art AI models can effectively aid in complex task comprehension and reward structure formulation. The framework is introduced in Appendix. Due to the presence of hemomorphic or isomorphic reward devices within semi-alignable Markov decision processes (MDPs) in section. 4.1, we demonstrate the efficacy of these machines in effectively translating rewards. Hemomorphic and isomorphic reward machines extract abstract alignment from semi-alignable MDPs across various domains. Utilizing these reward machines, seamless mapping between $\mathcal{U}, \mathcal{F}$, and $\mathcal{P}$ is achievable, allowing for state reward functions transfer from an original task's reward machine to a target task's reward machine, thereby promoting training efficiency. In section. 4.1, we introduce the basic setiing of reward machine, and in section. 4.1, we introduce the hemomorphic reward machine and isomorphic

reward machine to analyze the relationship of the reward machine for task with semi-alignable MDPs in different domains. Then we propose the reward transfer within different domain through reward machines.

## 4.1 REWARD MACHINE

Initially, we present the formulation of reward machines, which are designed to unveil the organization of non-Markovian reward functions related to tasks characterized by high-level events. Reward machines is typically defined as (Icarte et al., 2018):

**Definition 3.** *(Reward Machine). Considering a collection of propositional symbols $\mathcal{P}$, an assortment of (environment) states $\mathcal{S}$, and a range of actions $x$, a finite state machine reward machine (RM) constitutes a tuple $\mathcal{R}_{PSA} = < U, u_0, F, \mathcal{P}, \delta_u, \delta_r >$, with: $U \subseteq \mathcal{S}$ representing a limited set of states, $u_0 \in U$ denoting an initial state, $F$ defining a restricted set of terminal states ($F \cap U = \emptyset$), $\mathcal{P}$ signifying the set of propositional symbols, $\delta_u$ characterizing the reward machine state transition function $\delta_r : U \to [U \times P \to U]$ and $\delta_r$ embodying the state-reward function $\delta_r : U \to [U \times P \times U \to \mathbb{R}]$.*

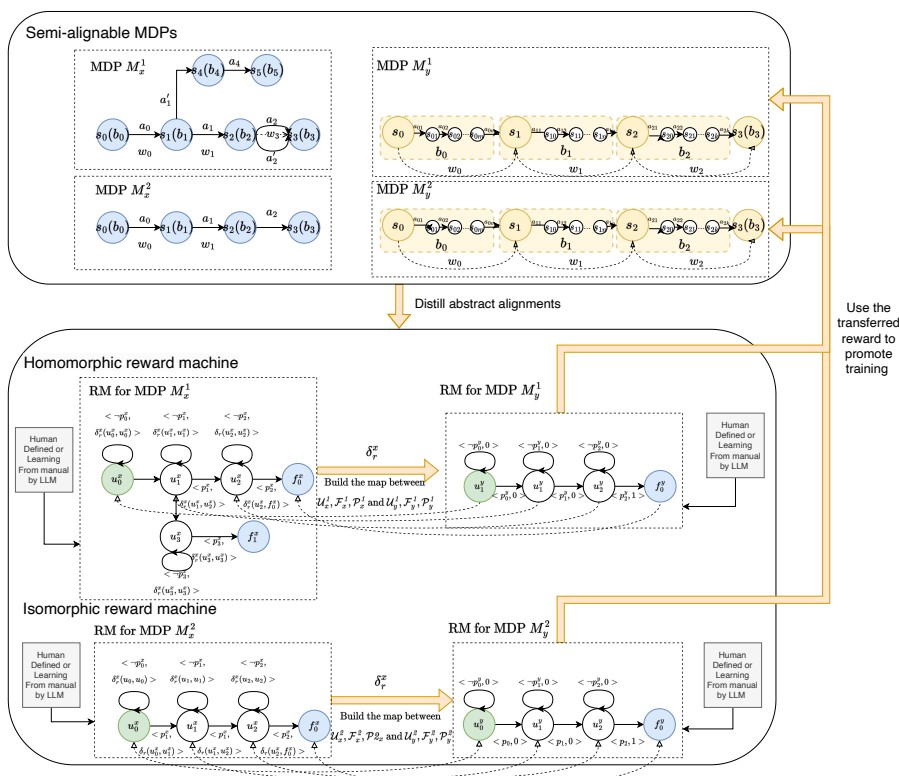

Figure 3: An overview of the Neural Reward Translation (NRT) framework. For cross-domain tasks in semi-alignable Markov Decision Processes (MDPs), a corresponding reward machine is produced using either human-defined criteria or language models to extract abstract alignments. Subsequently, a mapping between $\mathcal{U}$, $\mathcal{F}$, and $\mathcal{P}$ is established based on graph isomorphism and homomorphism theories. The state reward function is then transferred from the reward machine for the original task to the one for the target task, enabling the reuse of rewards to facilitate training in cross-domain tasks within semi-alignable MDPs.

Generally, using reward machines in reinforcement learning tasks requires extend the MDP.

**Definition 4.** *(MDP with Reward Machine). An MDP integrated with a Reward Machine is represented as a tuple $\mathcal{T}_{\mathcal{R}_{PSA}} = < S, x, p, \gamma, P, L, U, u_0, \delta_u, \delta_r >$, where $S, x, p, \gamma$ correspond to the state space, action space, transition function, and discount in the original MDP, while $P, U, u_0, \delta_u, \delta_r$ are determined by RM. Additionally, $L$ symbolizes a labelling function $L : S \to \mathcal{P}$.*

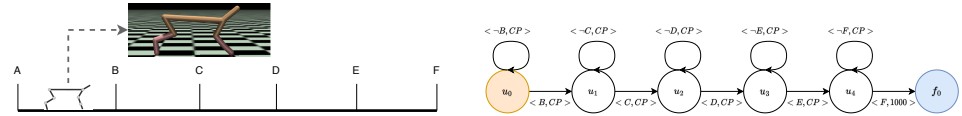

(a) HalfCheetah-v3 task. The target of the robot is to reach F in finite timesteps.

(b) Reward machine for HalfCheetah-v3 task. $f_0 \in F$ is the terminal state.

Figure 4: Example of reward machine for HalfCheetah-v3.

In every step, the agent in the environment perform action $a$ to move from $s$ to $s'$ in the MDP, the RM moves to state $u' = \delta_u(u, L(s'))$ and the agent receives reward $\hat{r}(s, a, s', u, u')$, for helping the reward transfer in our cross domain tasks, we design the reward function $\hat{r}$ as a combination of agent's original reward $r$ and the state reward function $\delta_r$ from reward machine as: At every stage, the agent situated within the environment executes action $a$, transitioning from state $s$ to $s'$ in the MDP. The RM shifts to state $u' = \delta_u(u, L(s'))$ from $u$, and the agent is granted reward $\hat{r}(s, a, s', u, u')$. To facilitate reward transfer across various domain tasks, we devise the reward function $\hat{r}$ as an amalgamation of the agent's inherent reward $r$ and the state reward function $\delta_r$ derived from the reward machine:

$$\hat{r}(s, a, s', u, u') = r(s, a, s') + \delta_r(u, u'), where\ u' = \delta_u(u, L(s')). \tag{1}$$

Eventually, the $\delta_r$ may be represented in terms of a potential-based reward shaping formulation:

$$\delta_u(u, u') = \gamma_{\mathcal{R}}\phi(u') - \phi(u), \text{where } \gamma_{\mathcal{R}} \text{ is the discount parameter.} \tag{2}$$

Furthermore, this study employs the expert of optimal value function for the agent in reinforcement learning to compute the potential value of the reward machine state as follows:

$$\phi(u) = \mathbb{E}_{s_i \sim S}[V^*(s_i, u_i | u_i = u)]. \tag{3}$$

Fig. 4 shows a reward machine example for the HalfCheetah-v3 task, where the robot aims to reach target $F$ in a limited number of steps. Unlike environment states ($s$) in reinforcement learning, reward machines have states ($u$) and transitions determined by $\mathcal{P}$. In Fig. 4(a), $\mathcal{P}$ depends on the robot's position relative to the target. The path is divided into six checkpoints—passing a checkpoint advances the reward machine state. Ultimately, reward machines deconstruct reinforcement learning tasks and facilitate a coarse-grained decomposition of reinforcement learning task structures, laying the groundwork for generalized cross-domain knowledge, as numerous tasks spanning diverse domains exhibit significant disparities in their fine-grained states, actions, and transitions. Then we introduce the concepts of hemomorphic and isomorphic reward machines to examine the relationships between reward machines and tasks within semi-alignable Markov Decision Processes (MDPs) in various domains. x connection between MDPs with reward machines and the extended MDP definition described in Section 3 can be observed. The abstract state $b \in y$ and the reward machine state $u \in U$ both represent a coarse-grained abstract structure of the reinforcement learning task. Consequently, we establish a bijection between the abstract state $b_i \in y$ and the corresponding reward machine state $u \in U$ (including terminal state $f \in F$): $b_i = \Theta(u_i)$, along with a bijection between abstract action state $w_i \in W_i$ and the corresponding reward machine events (propositional symbols) $p_i \in P$: $w_i = \Gamma(p_i)$. We then propose definitions for isomorphic and homomorphic reward machines, as depicted in Figure 3.

**Definition 5.** *(Isomorphic reward machine). Given two reward machines $\mathcal{R}_{PSA}^x$ and $\mathcal{R}_{PSA}^y$, $\mathcal{R}_{PSA}^x$ and $\mathcal{R}_{PSA}^y$ are isomorphic reward machines if and only if there exists bijection $h : \mathcal{P}^x \to \mathcal{P}^y$ and $g : \mathcal{U}^x \to \mathcal{U}^y$ such that $p_i^y = h(p_i^x), p_i^x = h^{-1}(p_i^y)$ and $u_i^y = g(u_i^x), u_i^x = g^{-1}(u_i^y)$.*

Given the bijections $b_i = \Theta(u_i)$ and $p_i \in P$: $w_i = \Gamma(p_i)$ introduced before, we can leverage Theorem 1 to describe the relationship amongst semi-alignable MDPs with isomorphic reward machines.

**Theorem 1.** *If two MDPs $M_x$ and $M_y$, $M_x \succeq M_y$ and $M_y \succeq M_x$, then their tasks have isomorphic reward machine $\mathcal{R}_{PSA}^x$ and $\mathcal{R}_{PSA}^y$.*

We prove the theorem in Appendix. For MDPs $M_x$ and $M_y$, if isomorphic reward machines exist for their tasks, the reward function of a reinforcement learning agent in $M_y$ can be represented as:

$$\hat{r}_{\delta_r^y}^y(s^y, a^y, s^{y\prime}, u^y, u^{y\prime}) = r^y(s^y, a^y, s^{y\prime}) + \delta_r^y(u^y, u^{y\prime}), \tag{4}$$

if $M_y$ uses the transferred $\delta_r^x$, then the reward function will be:

$$\hat{r}_{\delta_r^x}^y(s^y, a^y, s^{y'}, u^y, u^{y'}) = r^y(s^y, a^y, s^{y'}) + \delta_r^x(g(u^y), g(u^{y'})). \tag{5}$$

As both $\delta_r^x$ and $\delta_r^y$ employ potential-based reward shaping, we can deduce the following equation by examining the potential-based reward shaping properties:

$$\sum_{i=0}^{T-1} \hat{r}_{\delta_r^y}^y(s_i^y, a_i^y, s_{i+1}^y, u_i^y, u_{i+1}^y) = \sum_{i=0}^{T-1} \hat{r}_{\delta_r^x}^y(s_i^y, a_i^y, s_{i+1}^y, u_i^y, u_{i+1}^y), \tag{6}$$

where $T$ represents the task duration. The above is also established for transferring state reward functions from $M_y$ to $M_x$. Nevertheless, isomorphic reward machines impose a stringent condition that necessitates a bijection between semi-alignable MDPs' reward machines. We hence introduce homomorphic reward machines as a more relaxed formulation.

**Definition 6.** *(Homomorphic reward machine). Given two reward machine $\mathcal{R}_{PSA}^a$ and $\mathcal{R}_{PSA}^b$, $\mathcal{R}_{PSA}^a$ is $\mathcal{R}_{PSA}^b$ is Homomorphic reward machine if and only if there exists injection $h : \mathcal{P}^x \to \mathcal{P}^y$ and $g : \mathcal{U}^\S \to \mathcal{U}^\dagger$ let $p_i^y = h(p_i^x)$ and $u_i^y = g(u_i^x)$.*

Also given the bijections $b_i = \Theta(u_i)$ and $p_i \in P$: $w_i = \Gamma(p_i)$ introduced before, we can leverage Theorem 2 to describe the relationship amongst semi-alignable MDPs with homomorphic reward machines.

**Theorem 2.** *If two MDPs $M_x$ and $M_y$, $M_y \succeq M_x$ or $M_x \succeq M_y$, then their tasks have homomorphic reward machine $\mathcal{R}_{PSA}^x$ and $\mathcal{R}_{PSA}^y$.*

We have also proven the Theorem in Appendix. A.1. For $M_x$ and $M_y$ MDPs, if $M_x \succeq M_y$, then the reward function of a reinforcement learning agent in $M_y$ is denoted in the same manner as Equation (4). When $M_y$ implements the transferred state reward functions from $M_x$'s reward machine, the reward function remains consistent with Equation (5) while conserving the character described in Equation (6). Nevertheless, transferring rewards from $M_y$ to $M_x$ requires developing a piecewise function due to the unavailability of a bijection. The piece-wise function can be represented as follows:

$$\hat{r}_{\delta_r^y}^x(s^x, a^x, s^{x'}, u^x, u^{x'}) = r^x(s^x, a^x, s^{x'}) + \begin{cases} \delta_r^y(u^y, u^{y'}) \text{ if } u^x = g(u^y) \text{ and } u^{x'} = g(u^{y'}) \\ \delta_r^x(u^x, u^{x'}) \text{ otherwise} \end{cases} \tag{7}$$

As a result, a reward translation channel is established between semi-alignable MDPs. The reward function is transferred between the reward machines by first constructing corresponding reward machines manually or with LLM and then calculating the state reward function of the original task using Equation (3). Furthermore, the relationship between corresponding reward machines from the initial and target tasks must be constructed. Finally, rewards from different domain tasks are translated and reused in the target task, any reinforcement learning algorithm can directly use the translated reward and promote training.

## 5 EXPERIMENT

As reward machine can distill abstract alignment from semi-alignable MDPs within reinforcement learning tasks from different domains. This experiment aims to investigate how transferring rewards via isomorphic and homomorphic reward machines can enhance learning in reinforcement tasks across different domains by extracting abstract alignment from semi-alignable MDPs.

### 5.1 EVALUATION PROTOCOL

We evaluate the NRT framework within isomorphic and homomorphic reward machines using three experiments, where the original task is on the left and the target task is on the right. In each experiment, we train a reinforcement learning policy for the original task, calculate the state reward function $\delta_r$ according to Equation 3, and transfer it to the corresponding reward machine of the target task. Then we use reinforcement learning to train the target task within transferred reward, examine why the transferred reward translated from other domain can promote training. Detailed description of the environments and the reward machines are mentioned in Appendix.

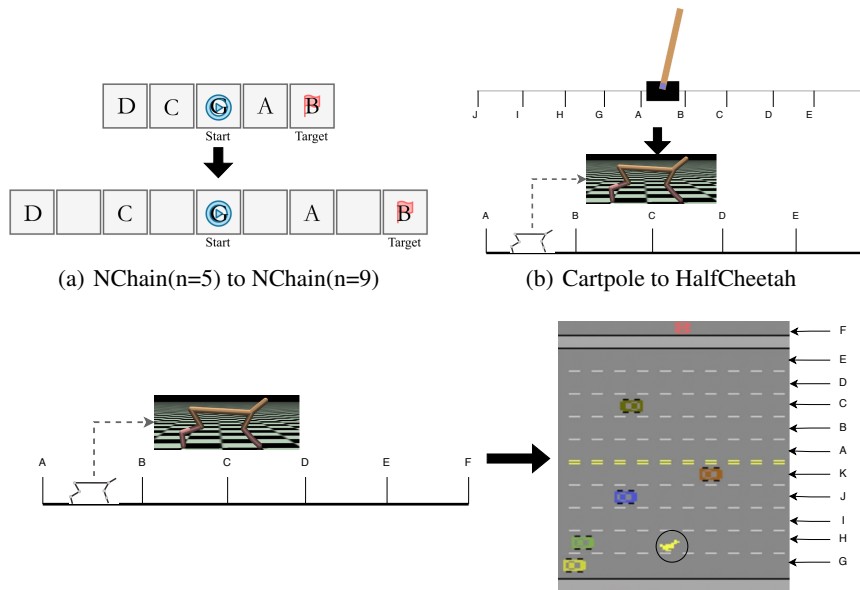

(a) NChain(n=5) to NChain(n=9)    (b) Cartpole to HalfCheetah

(c) HalfCheetah to Atari-Freeway

Figure 5: Three experiments we consider in this paper, NChain(n=5)-to-NChain(n=9) to examines NRT under isomorphic reward machines, Cartpole-to-HalfCheetah and HalfCheetah-to-Atari-Freeway examines NRT under homomorphic reward machines.

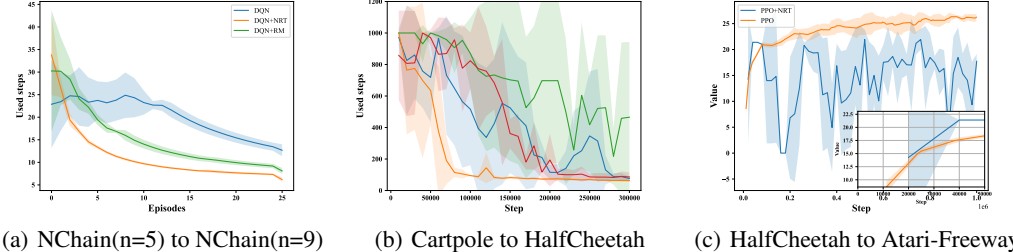

(a) NChain(n=5) to NChain(n=9)    (b) Cartpole to HalfCheetah    (c) HalfCheetah to Atari-Freeway

Figure 6: The learning curves for experiments. To provide a more intuitive demonstration of the number of steps the agent needs to take to reach the target, we use the agent's step numbers instead of accumulated rewards for HalfCheetah and NChain.

1. **NChain(n=5)-to-NChain(n=9)**: This experiment examines the effectiveness of the NRT framework under isomorphic reward machines by transferring state reward functions from NChain(n=5) to NChain(n=9). The NChain tasks involve chains of different lengths with sparse rewards given only when the agent successfully grabs a flag.

2. **Cartpole-to-HalfCheetah**: This experiment investigates the efficacy of the NRT framework for homomorphic reward machines by transferring state reward functions from the Cartpole task to the HalfCheetah task, which have been modified to hold an MDP relationship of $M_x \succeq M_y$. Both tasks employ the same settings as their respective versions in OpenAI-Gym (Brockman et al., 2016); however, their rewards are modified to be sparse, received only upon reaching specific goals.

3. **HalfCheetah-to-Atari-Freeway**: The third experiment further probes the NRT framework's performance under homomorphic reward machines by transferring state reward functions from the HalfCheetah task to the Atari-Freeway task, maintaining an MDP relationship of $M_y \succeq M_x$. The tasks adhere to the same settings as their respective OpenAI-Gym (Brockman et al., 2016) versions, with rewards granted depending on times of cross all lanes within 2048 timesteps.

## 5.2 RESULTS

Figure 6 presents a comparison of the training outcomes of NRT and baseline methods across three distinct experiments. Figure 6(a) illustrates the NChain(n=9) task training progression, utilizing a

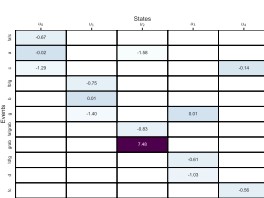 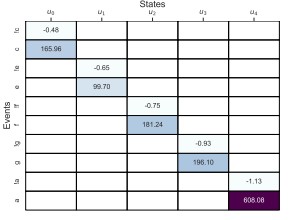 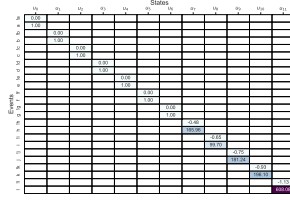

(a) transferred reward machine for the 9-Chain game
(b) transferred reward machine for the HalfCheetah-v3
(c) transferred reward machine for the Atari-Freeway

Figure 7: The transferred reward machines for experiments.

previously trained NChain(n=5) agent's state reward function due to their isomorphic reward machine structure. The experiment employed DQN as the baseline and DQN+RM (absent trained reward functions) for ablation analysis. The findings reveal that NRT significantly enhances training efficacy and performance in the NChain game. Additionally, Figure 7(a) displays the $\delta_r$ table within the NChain reward machine, highlighting the capacity of NRT to facilitate RL agent learning via transferred rewards between tasks within differing domains through isomorphic reward machines.

Figure 6(b) delineates the HalfCheetah task training process, deriving the state reward function from a trained Cartpole agent's homomorphic reward machine. Given the MDPs relationship $M_x \succeq M_y$, the reward function can be transmitted through corresponding reward machine mappings. DDPG was utilized as the baseline, while DDPG+RM (without a trained reward function) and DDPG+Reward (only accessing rewards without reward machine states) were subjected to ablation analysis. Results indicate NRT clearly improves training efficiency and performance in the HalfCheetah task. Figure 7(b) showcases the $\delta_r$ table for HalfCheetah's reward machine, providing direct guidance to reach objectives. Consequently, NRT fosters RL agent learning by transferring rewards through homomorphic reward machines across separate domains.

Finally, Figure 6(c) depicts the Atari-Freeway task training process, applying the state reward function obtained from a trained HalfCheetah agent's corresponding reward machine (here we directly employing the transferred function from Cartpole to HalfCheetah). Due to the MDPs relationship $M_y \succeq M_x$, Equation 7 is employed for transferring the state reward function. Figure 7(c) indicates that rewards seemingly guide agents toward their targets. However, disparities in task structure may lead to local optima entrapment upon reaching a positively reinforced event. Despite this limitation, the inter-domain transferred rewards without effective injection or bijection still offers limited initial guidance, as demonstrated by superior performance compared to the baseline within the first 50,000 steps. However, we have to admit NRT doesn't work good without injection or bijection.

The training results demonstrate that transferring rewards via isomorphic and homomorphic reward machines enhances learning in reinforcement tasks across different domains, improving training efficacy and performance, but may be limited in tasks without effective injection or bijection, potentially causing local optima entrapment.

## 6 CONCLUSION

In summary, this study presented the notion of semi-alignable MDPs to encourage broader domain knowledge transfer in cross-domain reinforcement learning, mirroring human learning mechanisms. x new framework, termed Neural Reward Translation (NRT), was introduced, utilizing reward machines for the abstraction and transmission of reward signals within a semi-alignable MDP context. NRT's performance was validated across various game environments, revealing its capacity to bolster training efficacy and flexibility for RL agents in diverse domains.

Nonetheless, certain constraints were identified, particularly with tasks lacking effective injection or bijection, where NRT might inadvertently lead to local optima entrapment. Additionally, constructing suitable reward machines and discerning relationships in more complex tasks remains challenging. Future research aims to explore further cross-domain situations wherein semi-alignable MDPs may offer valuable domain knowledge transfer, thereby expanding the applicability of NRT.

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
