# A   APPENDIX

## A.1   PROOF

**Theorem. 1** *If two MDPs $M_x$ and $M_y$, $M_x \succeq M_y$ and $M_y \succeq M_x$, then their tasks have isomorphic reward machine $\mathcal{R}^x_{PSA}$ and $\mathcal{R}^y_{PSA}$.*

*Proof. Theorem 1* can be proven by showing both necessity and sufficiency.

**1. Proof of necessity:** If $M_x \succeq M_y$ and $M_y \succeq M_x$, then their tasks have isomorphic reward machines.

Let there exist semi-reductions $r_{x \to y} = (\phi_{x \to y}, \psi_{x \to y})$ from $M_x$ to $M_y$ and $r_{y \to x} = (\phi_{x \to y}, \psi_{y \to x})$ from $M_y$ to $M_x$. According to Definition 1, these semi-reductions satisfy the $\pi^w$-optimality and $y$-dynamic conditions for all beliefs and world-states.

We now construct bijections $h : \mathcal{P}^x \to \mathcal{P}^y$ and $g : \mathcal{U}^{\S} \to \mathcal{U}^{\dagger}$ as follows:

$$h(p^x_i) = \Gamma^{-1}_y(\phi_{x \to y}(w^x_i)) = \Gamma^{-1}_y(w^y_i) = p^y_i,$$
$$g(u^x_i) = \Theta^{-1}_y(\psi_{x \to y}(b^x_i)) = \Theta^{-1}_y(b^y_i) = u^y_i.$$

Since $\phi_{x \to y}$ and $\psi_{x \to y}$ are functions, so are their inverses. Besides, and $\Gamma$ and $\Theta$ are bijections. According to the transitivity of bijections. These bijections also satisfy the conditions of the isomorphic reward machines in Theorem 5, as both semi-reductions preserve the $\pi^w$-optimality and $y$-dynamic properties.

**2. Proof of sufficiency:** If the tasks have isomorphic reward machines, then $M_x \succeq M_y$ and $M_y \succeq M_x$.

Given the isomorphic reward machines $\mathcal{R}^x_{PSA}$ and $\mathcal{R}^y_{PSA}$ with established bijections $h : \mathcal{P}^x \to \mathcal{P}^y$ and $g : \mathcal{U}^x \to \mathcal{U}^y$, we will prove that $M_x \succeq M_y$ and $M_y \succeq M_x$.

Define mapping functions $\phi_{x \to y}(w^x_i) = \Gamma_y(h(p^x_i))$ and $\psi_{x \to y}(b^x_i) = \Theta_y(g(u^x_i))$. Since the task structures are isomorphic, these mappings can be used to construct the semi-reduction $(\phi_{x \to y}, \psi_{x \to y})$ from $M_x$ to $M_y$ that satisfies the $\pi^w$-optimality and $y$-dynamic conditions in Definition 1, thus showing that $M_x \succeq M_y$.

Similarly, use the inverse of given bijections for the semi-reduction $r_{y \to x} = (\phi_{y \to x}, \psi_{y \to x})$. That is, set $\phi_{y \to x}(w^y_i) = \Gamma_x(h^{-1}(p^y_i)) = \Gamma_x(p^x_i) = w^x_i$ and $\psi_{y \to x}(b^y_i) = \Theta_x(g^{-1}(u^y_i)) = \Theta_x(u^x_i) = b^x_i$. Since $h$ and $g$ are bijections, their inverses exist and are also bijections. By using these mappings, we can construct a semi-reduction from $M_y$ to $M_x$ that satisfies the conditions in Definition 1, showing that $M_y \succeq M_x$.

Having proven both necessity and sufficiency, we conclude the proof of Theorem 1. $\square$

**Theorem. 2** *If two MDPs $M_x$ and $M_y$, $M_y \succeq M_x$ or $M_x \succeq M_y$, then their tasks have homomorphic reward machine $\mathcal{R}^x_{PSA}$ and $\mathcal{R}^y_{PSA}$.*

*Proof.* To prove Theorem 2, we need to show both necessity and sufficiency:

**1. Proof of necessity:** If $M_y \succeq M_x$ or $M_x \succeq M_y$, then their tasks have homomorphic reward machines.

Assume $M_x \succeq M_y$. Let there exist a semi-reduction $r_{x \to y} = (\phi_{x \to y}, \psi_{x \to y})$ from $M_x$ to $M_y$. According to Definition 1, these semi-reductions satisfy the $\pi^w$-optimality and $B$-dynamic conditions for all beliefs and world-states.

We now construct injection $h : \mathcal{P}^x \to \mathcal{P}^y$ and $g : \mathcal{U}^x \to \mathcal{U}^y$ as follows:

$$h(p^x_i) = \Gamma^{-1}_y(\phi_{x \to y}(w^x_i)) = \Gamma^{-1}_y(w^y_i) = p^y_i,$$
$$g(u^x_i) = \Theta^{-1}_y(\psi_{x \to y}(b^x_i)) = \Theta^{-1}_y(b^y_i) = u^y_i.$$

Since $\phi_{x \to y}$ and $\psi_{x \to y}$ are functions, so are their inverses. Besides, and $\Gamma$ and $\Theta$ are bijections. According to the transitivity of bijections. These bijections also satisfy the conditions the homomorphic reward machines in Theorem 6 because semi-reductions preserve the $\pi^w$-optimality and $B$-dynamic properties.

**2. Proof of sufficiency:** If the tasks have homomorphic reward machines, then $M_y \succeq M_x$ or $M_x \succeq M_y$.

Given the homomorphic reward machines $\mathcal{R}^x_{PSA}$ and $\mathcal{R}^y_{PSA}$ that satisfy the established injections $h : \mathcal{P}^x \to \mathcal{P}^y$ and $g : \mathcal{U}^x \to \mathcal{U}^y$, we will prove that either $M_y \succeq M_x$ or $M_x \succeq M_y$.

Assume without loss of generality that $|\mathcal{P}^y| \leq |\mathcal{P}^x|$ and $|\mathcal{U}^\dagger| \leq |\mathcal{U}^\S|$. Then we can define mapping functions $\phi_{x \to y}(w_i^x) = \Gamma_y(h(p_i^x)) = \Gamma_y(p_i^y) = w_i^y$ for every element in the domain of $h$. Similarly, define $\psi_{x \to y}(b_i^x) = \Theta_y(g(u_i^x)) = \Theta_y(u_i^y) = b_i^y$ for all elements in the domain of $g$

These mappings can be used to construct a semi-reduction $(\phi_{x \to y}, \psi_{x \to y})$ from $M_x$ to $M_y$ that satisfies the $\pi^w$-optimality and $B$-dynamic conditions in Definition 1. Thus, $M_x \succeq M_y$.

Having proven both necessity and sufficiency, we conclude that if two MDPs $M_x$ and $M_y$, $M_y \succeq M_x$ or $M_x \succeq M_y$, then their tasks have homomorphic reward machine $\mathcal{R}^x_{PSA}$ and $\mathcal{R}^y_{PSA}$. This completes the proof of Theorem 2. □

## A.2 LEARNING RM BY LLM

### A.2.1 FRAMEWORK

In this section, we present a methodology for reward machine learning, utilizing a large language model (LLM) informed by domain-specific knowledge, such as task manuals. As depicted in Figure 8, the language model is initially provided with a few-shot learning strategy to acquaint it with reward machine design principles. This comprises establishing the sets of propositional symbols $\mathcal{P}$ and reward machine states $\mathcal{U}$, devising an event extraction function for the environment incorporating the reward machine, and formulating the transition function $\delta_u$ along with the state reward function $\delta_r$.

Subsequently, the LLM is endowed with domain knowledge encompassing both reward machine definitions and environmental descriptions retrieved from the task manual. Afterward, user-generated queries are fielded by the LLM, which employs its acquired knowledge to respond and facilitate the construction of the reward machine. Given the intricate nature of reward machine development, we adopt a chain of thought framework to enhance the reasoning capabilities of the LLM.

### A.2.2 EXAMPLE OF LEARNING RM BY LLM

**Reward Machine Description:**

You are familiar with automata theory. A reward machine is defined as following:

Given a set of propositional symbols $\mathcal{P}$, a set of (environment) states $S$, and a set of actions $A$, a reward machine (RM) is a tuple $R_{PSA} = <U, u_0, F, \delta_u, \delta_r>$, where $U$ is a finite set of states, $u_0 \in U$ is an initial state, $F$ is a finite set of terminal states (where $U \cap F = \emptyset$, terminal states are not existed in $U$), $\delta_u$ is the state-transition function, $U \times 2^{\mathcal{P}} \to U \cup F$, $\delta_r$ is the reward-transition function, $U \to [U \times U \to \mathcal{R}]$

A reward machine $R_{PSA}$ starts in state $u_0$, and at each subsequent time is in some state $u_t \in U \cup F$. At every step $t$, the machine receives as input a truth assignment $\sigma_t$, which is a set that contains exactly those propositions in $\mathcal{P}$ that are currently true in the environment. For example, in an open door and get the key game, $\sigma_t = \{e\}$ if the agent opens the door $e$, while $\sigma_t = \{k\}$ if the agent gets the key $k$. Then the machine moves to the next state $u_{t+1} = \delta_u(u_t, \sigma_t)$ according to the state-transition function, and outputs a reward function $r_t = \delta_r(u_t)$ according to the state-reward function. This process repeats until the machine reaches a terminal state. Note that reward machines can model never-ending tasks by defining $F = \emptyset$.

**#Trail 1:HalfCheetah**

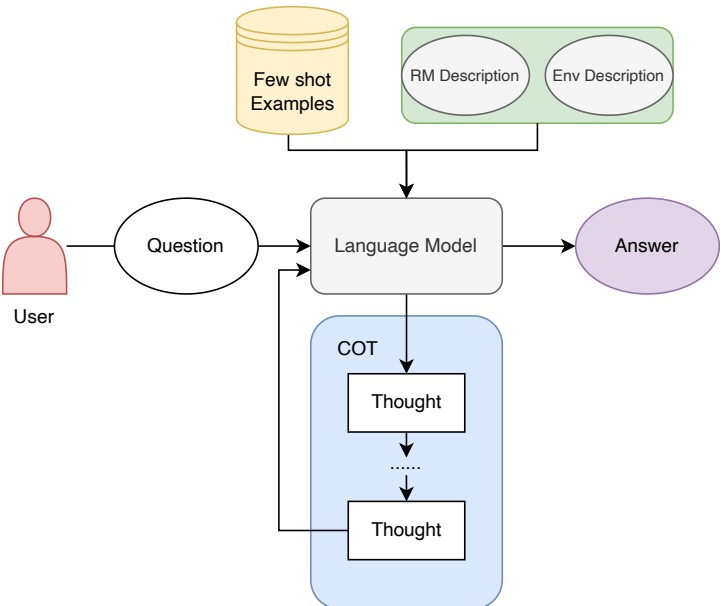

Figure 8: An overview of the framework to learn reward machine by large language model through domain knowledge.

**Environment Description:** HalfCheetah is part of the Mujoco environments. This environment is based on the work by P. Wawrzyński in "A Cat-Like Robot Real-Time Learning to Run". The HalfCheetah is a 2-dimensional robot consisting of 9 body parts and 8 joints connecting them (including two paws). The goal is to apply a torque on the joints to make the cheetah run forward (right) as fast as possible, with a positive reward allocated based on the distance moved forward and a negative reward allocated for moving backward. The torso and head of the cheetah are fixed, and the torque can only be applied to the other 6 joints over the front and back thighs (connecting to the torso), shins (connecting to the thighs), and feet (connecting to the shins). The robot needs to go through 5 checkpoints c(x_position=2), e(x_position=4), f(x_position=6), g(x_position=8), and a(x_position=10). The agent gets a reward of 1000 when it reaches a and the control penalization (CP) otherwise. Agent can get its current location by calling self.info['x_position'].

**Question**: Design the set of propositional symbols $\mathcal{P}$ for HalfCheetah environment.

**Think**: To design the set of propositional symbols $\mathcal{P}$, first I need to find the key events in the environment. The robot need to go through 6 checkpoints c(x_position=2), e(x_position=4), f(x_position=6), g(x_position=8), and a(x_position=10). I can use them as the propositional symbols, if the agent reach some checkpoint, it will reach a new reward machine state. Besides, I need to consider situations when agent don't reach the checkpoints.

**Answer**: The set of propositional symbols can be a set including checkpoints in HalfCheetah-v3 and situations not reaching the next checkpoints . $\mathcal{P} = \{c, !c, e, !e, f, !f, g, !g, a, !a\}$

**Question**: Design the get event function for the HalfCheetah with reward machine.

**Think**: Here events is depended on the propositional symbols, so the events including reach the checkpoints, and call self.info['x_position'] can get the agent's current x_position. Compare it with checkpoints' x_position, we can get the events.

**Answer**:

```
def get_events(self):
    events = ''
    if self.info['x_position'] < -10:
        events += 'b'
    if self.info['x_position'] > 10:
```

```
        events += 'a'
    if self.info['x_position'] < -2:
        events += 'd'
    if self.info['x_position'] > 2:
        events += 'c'
    if self.info['x_position'] > 4:
        events += 'e'
    if self.info['x_position'] > 6:
        events += 'f'
    if self.info['x_position'] > 8:
        events += 'g'
    return events
```

**Question:** Design the reward machine for the HalfCheetah environment.

**Think:** To design the reward machine. I need to verify the reward machine states first. As $\mathcal{P} = \{c, !c, e, !e, f, !f, g, !g, a, !a\}$. The reward machine including 5 states. As the agent reach checkpoint a, the game is finished. So these is one terminal state. Thus the reward machine state is $\{0, 1, 2, 3, 4, 5\}$ based on $\mathcal{P}$. Besides, as the agent gets a reward of 1000 when it reaches a and the control penalization (CP) otherwise. The $\delta_r$ can be designed based on it.

**Answer:**

$$0 \quad \text{\# initial state}$$
$$[5] \quad \text{\# terminal state}$$
$$(0, 0,' !c', \text{RewardControl}())$$
$$(0, 1,' c', \text{RewardControl}())$$
$$(1, 1,' !e', \text{RewardControl}())$$
$$(1, 2,' e', \text{RewardControl}())$$
$$(2, 2,' !f', \text{RewardControl}())$$
$$(2, 3,' f', \text{RewardControl}())$$
$$(3, 3,' !g', \text{RewardControl}())$$
$$(3, 4,' g', \text{RewardControl}())$$
$$(4, 4,' !a', \text{RewardControl}())$$
$$(4, 5,' a', \text{ConstantRewardFunction}(1000))$$

#### #Trail 2:Freeway

**Environment Description:** Freeway is part of the Atari environments. The objective in Freeway is to score as many points as you can by guiding your chicken across the ten lanes of traffic as many times as possible in limited steps. We use b-k to express the ten lanes, and a as the start point. Here is their coordinates: a(6), b(22), c(38), d(54), e(70), f(86), g(102), h(118), i(134), j(150), k(166). The agent gets a reward of 1000 when it crosses the ten lanes and 0 otherwise. After crossing the lanes, the agent will return to the start point and begin a new round.

**Question**: Design the set of propositional symbols $\mathcal{P}$ for Freeway environment.

**Think**: To design the set of propositional symbols $\mathcal{P}$, first I need to find the key events in the environment. The chicken has to cross ten lanes of traffic (b-k) and return to the start point a. I can use them as the propositional symbols if the agent reaches a lane or crosses all lanes and returns to the start point. Besides, I need to consider situations when the agent doesn't reach any checkpoints.

**Answer**: The set of propositional symbols can be a set including the start point and lanes in the Freeway environment and situations not reaching the next checkpoints.

$\mathcal{P} = \{a, !a, b, !b, c, !c, d, !d, e, !e, f, !f, g, !g, h, !h, i, !i, j, !j, k, !k\}$

**Question**: Design the get event function for the HalfCheetah with reward machine.

**Think**: Here events are dependent on the propositional symbols, so the events include reaching the start point or lanes. We can use the coordinates to detect the chicken's position. The chicken get agent_position x by calling self.info['x_position'], then compare it with the coordinates of start point and lanes.

**Answer**:

```
def get_events(self):
    events = ''
    x_position = self.info['x_position']

    if x_position == 6:
        events += 'a'
    elif x_position == 22:
        events += 'b'
    elif x_position == 38:
        events += 'c'
    elif x_position == 54:
        events += 'd'
    elif x_position == 70:
        events += 'e'
    elif x_position == 86:
        events += 'f'
    elif x_position == 102:
        events += 'g'
    elif x_position == 118:
        events += 'h'
    elif x_position == 134:
        events += 'i'
    elif x_position == 150:
        events += 'j'
    elif x_position == 166:
        events += 'k'

    return events
```

**Question**: Design the reward machine for the Freeway environment.

**Think**: To design the reward machine, I first need to identify the states, which correspond to the chicken's position (a-k) and inviable areas (!a-!k). There are 11 checkpoints (start point and lanes), and since the agent has to keep crossing lanes, there are no terminal states. The reward machine states should be numbered according to the propositional symbols, making the state set $\{0, 1, \ldots, 10\}$. Whenever the agent crosses all lanes and returns to the start point a, the reward is 1000, otherwise, the reward is 0.

**Answer**:

$$0 \quad \text{\# initial state}$$
$$[] \quad \text{\# terminal state}$$

$$(0, 0,' !a', \text{ConstantRewardFunction}(0))$$
$$(0, 1,' a', \text{ConstantRewardFunction}(0))$$
$$(1, 1,' !b', \text{ConstantRewardFunction}(0))$$
$$(1, 2,' b', \text{ConstantRewardFunction}(0))$$
$$(2, 2,' !c', \text{ConstantRewardFunction}(0))$$
$$(2, 3,' c', \text{ConstantRewardFunction}(0))$$
$$(3, 3,' !d', \text{ConstantRewardFunction}(0))$$
$$(3, 4,' d', \text{ConstantRewardFunction}(0))$$
$$(4, 4,' !e', \text{ConstantRewardFunction}(0))$$
$$(4, 5,' e', \text{ConstantRewardFunction}(0))$$
$$(5, 5,' !f', \text{ConstantRewardFunction}(0))$$
$$(5, 6,' f', \text{ConstantRewardFunction}(0))$$
$$(6, 6,' !g', \text{ConstantRewardFunction}(0))$$
$$(6, 7,' g', \text{ConstantRewardFunction}(0))$$
$$(7, 7,' !h', \text{ConstantRewardFunction}(0))$$
$$(7, 8,' h', \text{ConstantRewardFunction}(0))$$
$$(8, 8,' !i', \text{ConstantRewardFunction}(0))$$
$$(8, 9,' i', \text{ConstantRewardFunction}(0))$$
$$(9, 9,' !j', \text{ConstantRewardFunction}(0))$$
$$(9, 10,' j', \text{ConstantRewardFunction}(0))$$
$$(10, 10,' !k', \text{ConstantRewardFunction}(0))$$
$$(10, 0,' k', \text{ConstantRewardFunction}(1000))$$

We further visualize the generated reward machine from LLM in Figure. 9. Compared with the hand defined reward machine shown in Figure. 11 and Figure. 12, LLM has shown a strong ability to uncover task structure of reinforcement leanring.

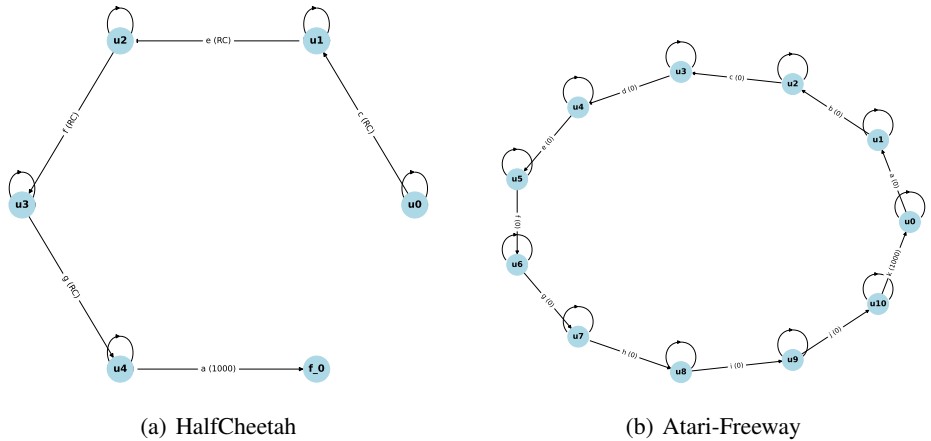

(a) HalfCheetah         (b) Atari-Freeway

Figure 9: The generated reward machine by LLM.

### A.3 EXPERIMENT DETAILS

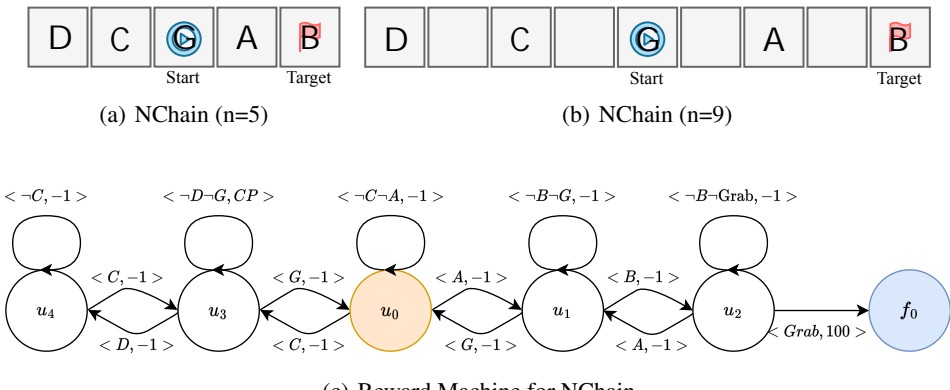

(a) NChain (n=5)                    (b) NChain (n=9)

(c) Reward Machine for NChain

Figure 10: NChain(n=5) to NChain(n=9). Their corresponding reward machines are isomorphic as they can use the same reward machine.

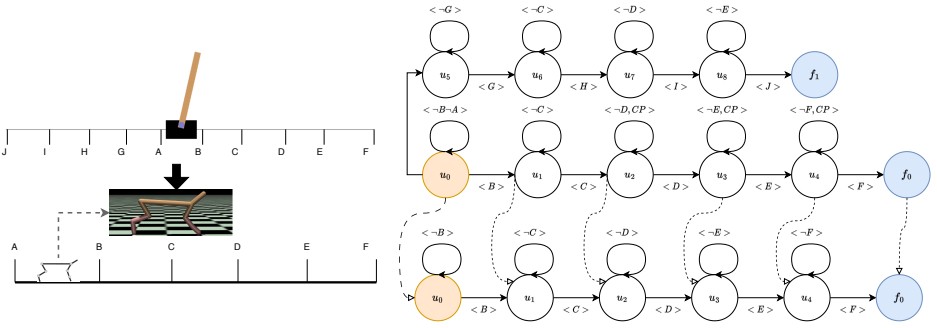

Figure 11: Cartpole to Halfcheetah. Their corresponding reward machine is homomrphic. And MDPs $M_x$ from Cartpole and MDPs $M_y$ from Halfcheetah hold $M_x \succeq M_y$.

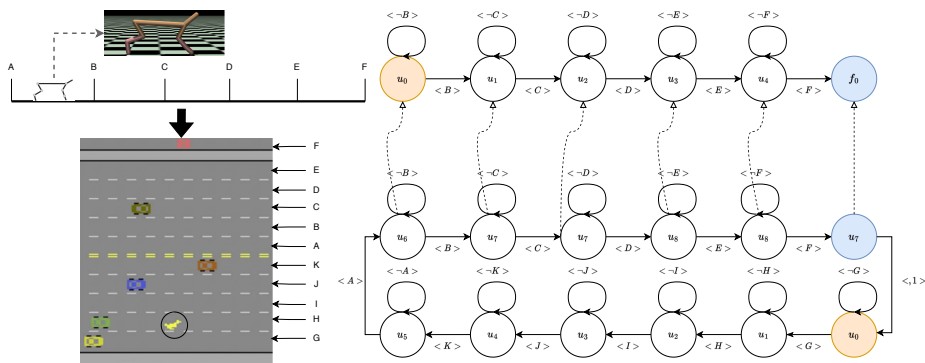

Figure 12: HalfCheetah to Atari-Freeway. Their corresponding reward machine is also homomrphic. But MDPs $M_x$ from HalfCheetah and MDPs $M_y$ from Atari-Freeway hold $M_y \succeq M_x$.

**NChain(n=5) to NChain(n=9)**: which has been shown in Figure. 10, where the state reward function transferred from NChain(n=5) to NChain(n=9) to examine the effectiveness of NRT under isomorphic reward machines. In NChain, where $n$ is the length of the chain, and the agent starts in the middle and can choose among the actions $\mathbb{A} = \{\text{left,right,grab}\}$, where grab does not change the state, but let the agent grab the provided flag if the agent is located at the position of the flag. The

state is the current coordinates of the agent. The reward is sparse as the reward is $0$ except the agent successfully grabs the flag. If the agent grab the flag, it will get the reward as $\frac{10 \times (1 - N_{steps})}{N_{max}}$. where $N_{steps}$ means the used step numbers for the agent to reach and grab the flag, and $N_{max}$ is the max horizon which is 20 for 5-Chain and 40 for 9-Chain games.

**Cartpole to HalfCheetah**: which has been shown in Figure. 4, where the state reward function transferred from Cartpole to HalfCheetah to examine the effectiveness of NRT under homomrphic reward machines as MDPs $M_x$ from Cartpole and MDPs $M_y$ from Atari-Freeway hold $M_x \succeq M_y$. For Cartpole, the action and state setting is the same as Cartpole-V3 in Openai-Gym (Brockman et al., 2016), and we change the reward as if the car reach the edge of the screem, it will get the reward as 1000. HalfCheetah use the same setting as HalfCheetah-V3 in Openai-Gym (Brockman et al., 2016).

**HalfCheetah to Atari-Freeway**: which has been shown in Figure. 1(b), where the state reward function transferred from HalfCheetah to Atari-Freeway to further examine the effectiveness of NRT under homomrphic reward machines as MDPs $M_x$ from HalfCheetah and MDPs $M_y$ from Atari-Freeway hold $M_y \succeq M_x$. HalfCheetah holds the same setting as before. For Atari-Freeway, the setting is the same as FreewayDeterministic-v4 in Openai-Gym (Brockman et al., 2016), where the agent will get reward $r = 1$ when crossing all the lanes of traffic. The target is to crossing all lanes as much as possible within 2048 steps.