# OpenReview forum: "Reward Translation via Reward Machine in Semi-Alignable MDPs"
_ICLR.cc/2024/Conference — Submitted to ICLR 2024_

### Official Review · Reviewer_FyDb · 2023-10-30

**Soundness:** 2 fair
**Presentation:** 2 fair
**Contribution:** 2 fair
**Rating:** 3
**Confidence:** 4

**Summary:**

This paper introduces a novel approach to enhancing deep reinforcement learning by addressing the challenge of reusing rewards across diverse domains. It presents the concept of semi-alignable Markov Decision Processes (MDPs) to uncover latent structural similarities between tasks in different domains and introduces the Neural Reward Translation (NRT) framework to facilitate reward transfer in semi-alignable MDPs. The paper demonstrates the effectiveness of this approach in various semi-alignable environments and provides a solution to the human engineering effort required for crafting reward machines. The work aims to improve learning efficiency in reinforcement learning by enabling the transfer of abstract skills and reward signals across disparate domains.

**Strengths:**

The paper's strengths lie in its innovative approach to addressing the challenge of reusing rewards in deep reinforcement learning across diverse domains. The introduction of semi-alignable Markov Decision Processes (MDPs) and the Neural Reward Translation (NRT) framework provides a novel foundation for identifying latent structural similarities, abstracting skills, and transferring rewards across different tasks.

**Weaknesses:**

- Figure 2 is visually hard to go through. Function definitions in the right subfigure (Semi-alignable MDPs) should be written in different lines: M_x >= M_y f:B_x -> B_y .... Also, figure 2 is referenced twice in the paper, and it's not thoroughly explained by the references of what is being shown, especially for the "Semi-alignable MDPs".

- First paragraph of section 3 should be grammatically revised. For example, in line 3: "shown in Figure. 2." -> "shown in Figure 2."

- What are the lines in Figure 6b? Legend is missing.

Overall, I believe the paper could benefit from another iteration of revision and improvement. Thus, I'm more inclined towards rejecting the paper.

**Questions:**

See above.

---

> ### Author Response · Authors · 2023-11-21
>
> Dear Reviewer FyDb,
>
> We appreciate your review and have addressed the explanations of semi-alignable MDPs and the missing legends in the commonly mentioned points section.

---

> > ### Comment · Reviewer_FyDb · 2023-11-22
> > **Response to authors**
> >
> > I would like to thank the authors for all their efforts to address the reviewers' concerns. However, after going through the discussion between the authors and other reviewers, I intend to keep my ratings as is. I believe the paper could massively benefit from another round of revision.

---

### Official Review · Reviewer_YSPR · 2023-10-30

**Soundness:** 1 poor
**Presentation:** 1 poor
**Contribution:** 2 fair
**Rating:** 3
**Confidence:** 3

**Summary:**

This work proposes the framework of semi-alignable MDPs as well as a procedure that aims to solve the problem of transferring knowledge between two semi-alignable MDPs referred to as Neural Reward Translation. The paper introduces the framework of a reward machine and subsequently derives semi-alignable MDPs as an alternative to standard alignable MDPs. Then, the framework of Neural Reward Translation is introduced in which reward machines are used to transfer reward functions across MDPs.The work concludes with an experimental evaluation of the suggested framework.

**Strengths:**

Motivation
* The motivation of the work is well established and clear. The idea of understanding how certain MDPs relate and extracting information from one MDP to learn another seems quite intriguing. Building frameworks that allow for the inclusion of large language models is also a promising direction as it might allow to automate the definition of reward translations.

Contextualization with prior work
I am not familiar with this type of literature but I am familiar with the standard reinforcement learning literature.
* It seems that the related work section as well as section 2.2 on reward machines cover a good number of related work to contextualize the present manuscript.

Novelty
* It seems that the idea of automating the understanding of MDP relationships via abstract concepts is relatively novel. However, I have to mention again that I am not very familiar with the sub-field literature.

**Weaknesses:**

Textual clarity
* The paper would benefit from additional proof reading as there are various typos and grammatical errors.
* The introduction contradicts itself with respect to prior work and it is not clear to me what the claim with respect to previous work’s capabilities is. It first states that Kim et al, Raychaudhuri et al provide methods that can handle unaligned demonstrations but then claims that the example with unaligned trajectories cannot be handled. See Q1.
* It is not clear from Figure 1 what an semi-alignable MDP exactly is.
* The reward machine is first mentioned very early on in the paper but not defined until section 4 which makes it very hard to follow much of the text. The section on reward machines did not explain properly what a reward machine is, what its inputs and outputs are or what its purpose is.
* Several references to the Appendix are missing the exact location of where to look, see e.g. P4 section 4.
* The construction of propositional symbols is hinted at several times but never explained. It is not clear how an LLM can be used to extract these symbols.
* Figures 2 and 3 are hard to read due to very small font size and very overloaded. It is hard to understand what they are supposed to demonstrate.

Mathematical rigor and clarity
Various mathematical concepts and definitions are unclear or used incorrectly, here are some examples:
* Section 3 introduces various new sets of variables that are not standard to the MDP. However, their motivation or meaning is not explained. It is not clear to me what an abstract state space or a skill space are. Then, the MDP does not contain the action and abstract state spaces anymore but rather undefined variables $A_y$ and $B_y$. The second task MDP also uses the same notation as the abstract space as an indicator. My guess is that this is a mix-up but it happens several times throughout the paper making it hard to follow which notation means what. Also, the notation $y$ is overloaded here since it refers to the abstract state space already.
* Definition 1 is not very clear. The notation $O$ is not defined and as a result I’m not sure what exactly the constraints on policy optimality and y-dynamics are. See Q2.
* Various instances in the paper use notation that is not explained or only explained much later. See e.g., page 4 section 4 “seamless mapping between $\mathcal{U}$, $\mathcal{F}$, and $\mathcal{P}$ is achievable”. I’m assuming that $\mathcal{U}$ is supposed to be $U$ and $\mathcal{F}$ is supposed to be $F$.
* There seems to be a typo for $\delta_u$ in Definition 3.
* The explanation of the function of a reward machine MDP in section 4.1 uses various definitions incorrectly or uses functions that are not defined. For instance, $\delta_u$ was defined as a function of limited states but now takes in a limited state as well as a propositional symbol, $\delta_r$ does not take two limited states, the MDP with a reward machine does not have a reward function $r$ or $\hat{r}$.
* The function $V$ is not defined in equation 3. My guess is that it is supposed to be the standard value function from the RL framework but then it is used incorrectly since it should be a function of a single state.

Since the notation is used incorrectly in various places or not defined, I did not check the proofs for correctness.

Experimental evaluation
* The environment descriptions are very vague. In section 5.1, I’m not sure what it means that an original task is on the left and a target task on the right. From section 5, It is not clear to me what the experimental setup is. Equation 3 does not compute a reward function.
* Figure 5b is missing a legend and I cannot assume that the colors correspond to the same baselines as in the other Figures since there is a red line not present in the other plots and the ordering of NRT changes across plots. As a result, I cannot determine whether the claims with respect to this experiment are accurate.
* The experimental evaluation is rather short and only contains very simple environments and the conclusions seem either hard to determine due to high variance (Fig 6b) or weak results (Fig 6c).
* One of the main claims of the experiments is that “The training results demonstrate that transferring rewards via isomorphic and homomorphic reward machines enhances learning in reinforcement tasks across different domains, improving training efficacy and performance.” However, this claim is not well supported since isomorphic reward functions have not been tested and homomorphic reward functions only worked in a very limited set of the experiments.

Overall, I believe this paper would benefit from iterations of clarity improvements both on the textual as well as mathematical front. The experiments could be more extensive and some of the claims seem overstated. As a result, I recommend rejection of the manuscript.

**Questions:**

Q1. Can you elaborate on why exactly previous methods are unable to align the presented MDPs in Figure 1? It is not clear to me what properties semi-alignable MDPs would have and how I would identify them. Can you explain how we can identify a semi-alignable MDP?

Q2. In Definition 1, the text uses an O notation that is not explained. Can you elaborate on what exactly this means?

Q3. Can you clarify the experimental setup?

Q4. Can you clarify what the purpose of the propositional symbols is and how they determine mappings? Maybe with an example?

---

> ### Author Response · Authors · 2023-11-21
>
> Dear Reviewer YSPR,
>
> Thank you for your valuable input. We have addressed the experimental setup and the differences between alignable MDPs and semi-alignable MDPs in the commonly mentioned points section.
>
> `Q1: The test uses an O notation that is not explained.`
>
> We sincerely apologize for any confusion this might have caused. In this context, "O" represents the optimality function concerning abstract actions and abstract states. This means that agents within both environments can learn optimal policies that can be mapped.
>
> `Q2: Can you clarify what the purpose of the propositional symbols is and how they determine mappings? Maybe with an example?`
>
> Propositional symbols serve as checkpoints within the environment and are often defined in the task description. As an example, in a game where the objective is to find a key and open a door, the key and the door would represent the propositional symbols.
>
> `Q3: The training results demonstrate that transferring rewards via isomorphic and homomorphic reward machines enhances learning in reinforcement tasks across different domains, improving training efficacy and performance.`
>
> Our experiments with NChain(n=5) to NChain(n=9) showcase the performance of isomorphic reward machines, since both environments share the same reward machine. On the other hand, our Cartpole to Halfcheetah experiment demonstrates the performance of homomorphic reward machines. The structure of these reward machines can be found in the Appendix for comparison with their respective definitions. However, we must acknowledge that our experiments' scale may not make them entirely persuasive, and we will strive to enhance their robustness.
>
> `Q4: Definition of $\mathcal{F}$ and $\mathcal{U}$.`
>
> We apologize if this caused any confusion. Both $\mathcal{F}$ and $\mathcal{U}$ are components of the reward machine. While $\mathcal{F}$ refers to the set of terminal states where the reward machine is done (and subsequently, the reinforcement learning task is complete for that episode), $\mathcal{U}$ denotes a set of general states, such as "The agent has found the key" in the "Find the key and open the door" game example.

---

> > ### Comment · Reviewer_YSPR · 2023-11-22
> >
> > Thank you for the clarification and responses to my questions. I have also read other reviews and corresponding responses.
> >
> > It seems that no updated manuscript was submitted and most of the weaknesses I outlined in my initial review remain unaddressed. As a result, I will retain my score and favor rejection.

---

### Official Review · Reviewer_LeVr · 2023-10-31

**Soundness:** 2 fair
**Presentation:** 3 good
**Contribution:** 2 fair
**Rating:** 6
**Confidence:** 3

**Summary:**

This paper propose a definition for semi-alignable MDPs and the corresponding Neural Reward Translation Model. The problem this paper investigated is cross-domain transfer learning via reward machine transfer. Experiments show that reward transfer between different environments are effective. In detail, the reward model in one domain can be learnt to map to another domain with LLMs labeling or human labeling. Therefore, different domain tasks can be aligned.

**Strengths:**

This method is novel and easy to follow. Aligning different tasks with domain shifts are important to imitation learning community. I think this method gives us a new solution by mapping the rewards rather than mapping the observation. The key by doing that is that LLMs can be easily obtained to label the rewards for different proxy task behaviors. As a consequence, this method is built on the success of LLMs. I think it is a good and interesting work for other researchers to follow.

**Weaknesses:**

1. I would like to see some results evaluated by the ground truth episode rewards.
2. I think this setting is more like imitation learning setting. There could be some expert demos for aligning different tasks. The expert demo can show the task behavior much more clear. However, this paper seems to use proxy task data to label the rewards. Therefore, the NRT could be built on this set of data.

**Questions:**

I am confused of how much data should we use to align different domain tasks?

The format should be revised such as Figure 5-6.

---

> ### Author Response · Authors · 2023-11-20
>
> Dear Reviewer LeVr,
>
> Thank you for your valuable feedback and your encouragement regarding our work. We plan to refine our paper according to your suggestions and provide answers to your additional queries.
>
> `Q1: How much data should we use to align different domain tasks?`
>
> In NRT, we either utilize GPT or create hand-defined reward machines based on task manuals and environment descriptions. Consequently, we don't need to access expert demonstrations like those used in imitation learning or inverse reinforcement learning.

---

> > ### Comment · Reviewer_LeVr · 2023-11-23
> > **Thanks for your response**
> >
> > Thanks for your response. I would like to keep my score.

---

### Official Review · Reviewer_heMd · 2023-11-03

**Soundness:** 2 fair
**Presentation:** 1 poor
**Contribution:** 2 fair
**Rating:** 3
**Confidence:** 2

**Summary:**

This paper introduces a new concept of semi-alignable MDP that requires more relaxed alignments between two MDPs compared to alignable MDPs considered in prior work for transferring the reward across different tasks and environments. For reward transfer, the paper introduces Neural Reward Translation that uses reward machines. The proposed method is evaluated in three setups (shorter NChain to longer NChain, Cartpole -> Halfcheetah, and HalfCheetah to Atari-Freeway).

**Strengths:**

- The paper tackles an important problem of transferring rewards, which would allow for reducing the significant cost of designing dense rewards.
- Idea to use LLM to construct a reward machine is interesting.
- The paper shows that the proposed transfer mechanism can improve the performance on several tasks.

**Weaknesses:**

- I don't think the quality of writing should be an important factor in assessing the paper, but it's problematic when it actually makes it difficult to understand the main point of the paper. It's extremely difficult to read and parse the paper because of a lot of formatting errors, typos, and the lack of effort in organizing the contents. I had to guess the missing parts of the sentences multiple times while reading the paper, and the method section just dumps everything without trying to emphasizing what's the main content the paper is aiming to deliver. It's difficult to recommend the paper to be accepted at this status, and it needs a significant amount of revision to reach the quality of writing required for a conference like ICLR.
- The intuitive motivation in Figure 1 (Cartpole -> HalfCheetah) is very confusing, as the goal of balancing the cartpole is significantly different from the goal of halfcheetah that makes it run faster as far as possible. Giving a more intuitive example or supporting the argument here could be useful for improving the clarify of the paper.
- The usage of GPT-4 for constructing a reward machine is interesting, but if it's possible, it's not clear to me why it's still necessary to transfer the reward, because it might be possible to generate the rewards for the downstream task directly. This should be thoroughly investigated by including an additional baseline.

**Questions:**

- It seems like the paper is directly referring some works in the category of imitation learning as RL, it could be nice to be more formal in this.
- Please add . in the abstract between two sentence: varying domains Subsequently,
- In abstract, please make it clear that semi-alignable MDPs are new concepts introduced in the paper.
- In page 2, `Servel -> Several`
- Formatting for the figure captions is broken. Please fix this.
- In page 4, the following sentence is not complete: `However, direct finding the semi-reduction between M T x and M T y because the abstract state space y and the skill space W in both domains are indeterminate.`
- Please re-organize the method section, instead of having one subsection, and please avoid dumping everything with consecutive sequences.
- It's difficult to parse the following sentence: `x connection between MDPs with reward machines and the extended MDP definition described in Section 3 can be observed.`
- Please try to incorporate the contents in Appendix that describe how you generated reward machines with LLMs.
- There's no legend in Figure 6(b), so that I can't know which line corresponds to which variants.

---

> ### Author Response · Authors · 2023-11-20
>
> Dear Reviewer Hemd,
>
> Thank you for your insightful review and your kind encouragement regarding our work. We acknowledge that our paper was written in haste and agree that some mistakes have been made. We appreciate your advice and will make the necessary modifications to our paper.
>
> `Q1: The intuitive motivation in Figure 1.`
>
> As explained in the commonly mentioned questions, we've adjusted the cartpole's target to reach the edge of the screen. In this new setting, we want to illustrate that a direct mapping doesn't exist between state and action spaces or transition functions in the two environments, which means they cannot be characterized as an alignable MDP and as defined by Kim, et al. The NChain might serve as a better example where the NChain(n=5) and NChain(n=9) share the same goal, but their different horizon lengths prevent them from ...
>
> `Q2: The necessity of using GPT-4 to build the reward machine rather than directly learning the reward.`
>
> We recognize that utilizing LLMs to learn rewards for reinforcement learning is a fascinating domain, although it extends beyond our current objectives. Our primary intention in this paper is to construct a bridge allowing semi-alignment between MDPs in different environments, enabling the reuse or transfer of rewards acquired from trained environments without the direct learning of rewards. Reward Machines serve as the bridge we employ. Ultimately, our experiments involve both hand-defined reward machines and LLM-generated reward machines, and we've found that LLMs can effectively learn a similar reward machine based on the task manual.

---

> > ### Comment · Reviewer_heMd · 2023-11-23
> >
> > Thanks for the response and clarification. But I remain my score as the draft is not updated yet, and agree with other reviewers that this paper could improve a lot by having one more round of revision.

---

### Author Response · Authors · 2023-11-20

Thank you for your thoughtful reviews. I would like to acknowledge that the paper was written under some time constraints, which may have resulted in confusing descriptions. In response to commonly mentioned questions, please find clarifications below.

`Q1: Differences between semi-alignable MDPs and alignable MDPs.`

The introduction of semi-alignable MDPs is intended to describe a relationship between MDPs where there is no direct map between their transition functions or states and actions. In contrast, alignable MDP theory aims to provide a theoretical basis to learn alignment between unpaired, unaligned demonstrations. Semi-alignable MDPs can be considered as an extension of alignable MDPs, with alignment existing on abstract action (similar to the options in hierarchical reinforcement learning) and abstract state (the state of the reward machine). Figure 2 provides an intuitive representation of their differences.

`Q2: The missing legend in Figure 5b.`

Apologies for this oversight. In this figure, the green line corresponds to DDPG performance, the blue line represents DDPG+RM, the red line indicates DDPG+Reward, and the brown line denotes DDPG+NRT.

`Q3: The experimental settings.`

Due to space limitations, we did not describe the experimental settings thoroughly. We conducted three experiments including NChain(n=5) to NChain(n=9), Cartpole-to-HalfCheetah, and HalfCheetah-to-Atari-Freeway. The left environments are source environments, and those on the right are target environments. For the Cartpole environment, we modified its objective to reach the edge of the screen and provided a sparse reward of 1000. To demonstrate NRT performance, we built reward machines for each environment, with mappings of reward machine states shown in the Appendix.

---

### Meta-Review · Area_Chair_q2Gc · 2023-12-12

**Metareview:**

### Summary
This paper proposes a method to improve deep reinforcement learning by tackling the challenge of reusing rewards across diverse domains. It introduces semi-alignable Markov Decision Processes (MDPs) to uncover structural similarities among tasks in different domains. The proposed Neural Reward Translation (NRT) framework enables reward transfer in semi-alignable MDPs, facilitating reward reuse in reinforcement learning across varied domains. Demonstrating its efficacy in multiple environments, this approach minimizes the human effort needed for crafting reward machines, aiming to enhance learning efficiency by enabling the transfer of abstract skills and rewards across disparate domains.

### Decision

The proposed idea is interesting and aims to address an important issue. Nevertheless, the reviewers have identified several common weaknesses in the paper, which I can help summarize and group together for the meta-review:

- **Writing Quality and Clarity**: Multiple reviewers noted serious issues with the clarity and quality of writing. There are formatting errors, typos, and a lack of coherent organization. Sentences are incomplete or difficult to understand, making it challenging to grasp the paper's main points. Revision for substantial improvements in writing quality is needed.

- **Lack of Clarity in Examples and Explanation:** The intuitive motivation provided in Figure 1 was confusing. Reviewers suggest using a clearer example or providing better support to enhance the paper's clarity regarding the transition between different tasks.

- **Explanation of Methodology and Necessity:** GPT-4 for constructing a reward machine needs clearer justification. Reviewers question the necessity of transferring the reward and suggest investigating if generating rewards directly for the downstream task is feasible. They call for additional baselines to support this.

- **Insufficient Experimental Evaluation:** Reviewers express dissatisfaction with the experimental evaluation. They find it lacking in persuasiveness and clarity, requiring more convincing results to support the paper's claims.

- **Inadequate Figures and Visual Presentation:** Figure 2 and other figures are visually challenging to comprehend. There are issues with legends, formatting, and explanations. The paper references figures without thoroughly explaining their content, causing confusion.

- **Grammar and Presentation Errors:** The paper has grammatical issues, requiring revision for better readability and clarity. Specific paragraphs and lines need grammatical revision for improved presentation.

The recommendation from the reviewers appears that the paper needs substantial revisions across multiple aspects, including writing quality, clarity in explanations and examples, better justification of methodology, improved experimental evaluation, enhanced visual presentation, and addressing grammar and alignment issues related to task setting and data usage. The collective feedback points towards the rejection of this paper. I would recommend the authors do a significant revision of this paper before submitting it to another venue.

**Justification For Why Not Higher Score:**

The reviewers collectively agreed to reject this paper.

**Justification For Why Not Lower Score:**

N/A

---

### Decision · Program_Chairs · 2024-01-16

Reject